# OpenReview forum: "Pruning Edges and Gradients to Learn Hypergraphs from Larger Sets"
_ICLR.cc/2022/Conference — ICLR 2022 Submitted_

### Official Review · Reviewer_am1Q · 2021-10-29

**Correctness:** 2
**Technical Novelty And Significance:** 2
**Empirical Novelty And Significance:** Not applicable
**Recommendation:** 5
**Confidence:** 3

**Main Review:**

Pro:
1. Predicting the set to hypergraph is a very difficult task as the solution space is O(2^n). This paper provides many tricks to increase the performance and decrease the complexity in training the models. For example, backprop with skip, recurrent training instead of stacking, reducing the solution space to O(kn).
2. This paper trains only on positive edges. They proved this loss is similar to the overall loss. This step could greatly increase the efficiency as reduced training space.
3. They benchmarked on three different tasks and showed a performance increase. The ablation study also revealed the proposed tricks could help training.

Con:
1. The writing and the structure of the paper are a little confusing. Though the author took lots of effort in introducing the background. But I still fail to understand what is the overall f function to construct the hypergraph? And since only trained on positive edges, how did the author select the predicted hyperedges from the overall k edges?
2. The proposed tricks are quite heuristic and do not provide theoretical insights for performance increase.
3. The introduction of the Hungarian algorithm in selecting the edges does not make sense to me. Though the orderless nature of the edge in the hypergraph is important, the introduction of the algorithm greatly limited the generalization of the methods. And bring bias in the performance, since the Hungarian algorithm selects the number of edges and best corresponding edges to train the model. When ground truth is not available, I don't know how the proposed algorithm will choose the number of edges and best edges to construct the hypergraph, i.e., how the model generalizes in a broader case rather than fitting the ground truth data. And I suspect the Hungarian algorithm put an unfair advantage of the proposed model in comparing with other methods. I would suggest the author remove the Hungarian algorithm and only try to derive the incidence matrix I with the proposed model.
4. Following comment 3, the author could introduce more detail about the input, output, prerequisites of the model. Specifically, the author could explain how they choose the number of edges, how to select predicted edges.









**Summary Of The Paper:**

This paper proposed an efficient algorithm to tackle the set to hypergraph problem by utilizing recurrent training, pruning negative edges and backprop with skips. It then benchmarked the performance with prior works on the particle partitioning, convex hall and delaunay triangulation tasks. The proposed method showed a performance increase over previous methods.



**Summary Of The Review:**

This paper proposes a sufficient method to predict hypergraphs from sets and showed performance increase. Currently, it lacks sufficient detail, and may require further evaluation to justify the claims.

---

### Official Review · Reviewer_ezwC · 2021-11-01

**Correctness:** 2
**Technical Novelty And Significance:** 2
**Empirical Novelty And Significance:** 3
**Recommendation:** 3
**Confidence:** 2

**Main Review:**

I find it hard to assess the contribution of this paper because I feel like I don't understand it. The English is good but there is a lack of definitions and explanations regarding the considered task, terminology, and mathematical notation, which makes it hard to make sense of the proposed method. In addition, the few mathematical definitions which are given do not seem to be complete. For example, the proposed loss function has a trivial solution and functions which are introduced later are not part of the loss (see comments below). Likewise, the experimental setting is not detailed. How the hyperparameters and the optimization parameters are tuned/set is not described for the proposed method and the competitors. As a result, I believe that the most actionable approach for the authors is that I list all the questions and issues I have in the following, to make clear where this paper needs improvement.

# Questions & Issues:
## Section 1 - Introduction:
* From what or on the basis of what is the hypergraph learned from the set? What exactly is the input?
* Based on what do you decide whether an edge exists or not?
* The terminology of negative edges being also called non-existent should be introduced before this terminology is used in this section.
## Section 2 - Scaling by pruning non-existing edges:
* Why do non-existing edges have to be pruned, why are they modeled in the first place?
* How is a hypergraph defined? How is a hyperedge defined?
* What is $V\subset \mathcal{V}$?
* What does it mean to "remove excess rows" from the incidence matrix? What is an excess row? I would expect that removing the row decreases the dimensionality of the incidence matrix. This is however not reflected in the mathematical notation.
* What does it mean to "supervise" an incidence $I_{ji}$?
* "a positive edge in the incidence matrix contains both zeros and ones" (-> what does that mean? Further:) "ensuring that the binary classifier sees both positive and negative examples"(->what is a pos/neg example?)
* "the order of the entries is fully decided by the order of the nodes" -> why, what does that mean?
* Mid Sec. 2 the ground truth appears and the loss is defined as sum of cross entropies $H(I_{i,j},I^*_{i,j})$. This loss function has a trivial minimum for $I=I^*$. I guess that some constraints are needed there. Also, where does the ground truth come from? Is this part of the input?
* "Since the losses are equivalent up to an additive constant, the gradients are exactly equal" -> I don't think that's true, $\nabla_I\mathcal{L}(I,I^*)\neq \nabla_J\mathcal{L}(J,J^*)$ already because $I$ and $J$ don't necessarily have the same dimensionality.

## Section 3 - Scaling by Pruning non-essential gradients
* Here, neural networks are mentioned for the first time, but the reader doesn't know here for what the NN is used and how it fits into the considered task.
* What is model $f$ representing? How do time steps fit into the considered task? Where is $f$ in the objective?
* What does it mean to supervise an iteration or step?
* In Alg. 1, the updates in the for $t$ loops do not depend on $t$.

## Section 4 - Scaling set-to-hypergraph prediction
* What is $x_i$?
* Introduce the abbrev. MLP, how is function $\texttt{MLP}$ defined?
* How is function $\texttt{DeepSets}$ defined?
* What is the existence indicator $\sigma$? How can Eq. (6) be seen as factorizing the probability?

## Section 5 - Experiments
* How are parameters tuned, and what are the optimization parameter settings (step-size etc.). How are parameters set for competitors?
* How is inference performed by the set-to-hypergraph models?
* How is the sd computed in Table 1? Are multiple runs compared or multiple datasets/test splits?
* How does the ML task to predict the convex hull differ from the one in computational geometry? Why is this task challenging for ML models but well understood in computational geometry?
* Why does "pruning the edges" improve the predictive performance?
* How is the number of iterations determined in the experiments?
* How does the proposed optimization relate to TBPTT? How can TBPTT and backprop with skips improve the performance in comparison to backprop? When would it not help to apply this?
* How does the run time of the proposed method relate to the one of competitors?

## Section 6 - Related work
* I would put this section earlier to introduce the competitors in the experiments.

## Section 7 - Conclusion and Future Work
* Nice, that you point out the limitations. But how does the input dimension exactly influence the feasibility of the proposed model?




**Summary Of The Paper:**

The authors propose a set-to-hypergraph model where the hypergraph is represented by an incidence matrix in contrast to the usually used adjacency tensors. The incidence matrix is an edge times node matrix, which reflects only existing edges and is hence suitable to reduce the memory requirement when the amount of existing edges is sparse. The authors propose for the optimization of the proposed model
 an SGD scheme where a batch update is performed on the hypergraph which is returned by a recurrent neural network after a varying amount of time steps.
The experiments compare the proposed approach with 3 competitors on four datasets.

**Summary Of The Review:**

Very unaccessible paper which might be understandable by experts in the field but not by a wider audience. Presentation needs to be improved, mathematical notation needs to be completed and experimental analysis needs to be clarified before I would consider this paper eligible for acceptance.

---

### Official Review · Reviewer_d3zQ · 2021-11-02

**Correctness:** 3
**Technical Novelty And Significance:** 3
**Empirical Novelty And Significance:** 2
**Recommendation:** 5
**Confidence:** 3

**Main Review:**

The set-to-hypergraph prediction is an interesting problem. The graph size is indeed the bottleneck of this problem. The paper focuses on this point and proposes its solutions. From the experimental results, the method is effective in performance. However, there are some critical defects in this paper.

1, The proof of Proposition 1 (Supervising positive edges only) should be the key to pruning negative edges. However, I find the first sentence of the proof in Appendix A is "For the sake of rigor, we first summarize the relevant assumptions from ??. ". This statement directly hinders me from understanding the whole proof. This kind of typo is fatal. I hope the authors can revise them in detail.

2, Some sentences are confusing. For example, on Page2 "if an edge connects every node in V ⊂ V then there exists a relation between the input elements {xi ∈ X|vi ∈ V }. ". I am confused about how an edge can connect every node? One edge at most connects two nodes, right?

3, The paper claims that specifying a maximum number of edges k is sufficiently large to cover all (or most) hypergraphs of interest. I think k should be a hyper-parameter and is very important. There is no content about how to determine and tune k, and the parameter analysis of k.

4, The core work of the paper is to scale the set-to-hypergraph prediction. I am curious about why there is no absolute training time comparison shown in the experimental section. The relative training time shown in Figure 2 is hard to understand for me.


**Summary Of The Paper:**

This paper improves the asymptotic scaling and enables the learning task to have higher-order edges by only representing and supervising a set of positive edges. In common benchmark tests, this paper has proved that the proposed method is superior to previous work while providing more favorable asymptotic scaling behavior. In further evaluation, this paper emphasizes the importance of repetition in solving the inherent complexity of the problem.

**Summary Of The Review:**

The paper works on a critical problem and proposes a reasonable solution. However, there are some fatal typos and confusing statements. The experimental section lacks some critical results like absolute running time and only tests on one dataset. The authors should solve my doubts above and modify the paper.

---

### Official Review · Reviewer_hMC3 · 2021-11-03

**Correctness:** 3
**Technical Novelty And Significance:** 2
**Empirical Novelty And Significance:** 3
**Recommendation:** 5
**Confidence:** 3

**Main Review:**

Strengths:
*  Proposition 1 is an interesting finding.

* There are plenty of experimental results to show the empirical advantages.

Weaknesses:
* The technical contributions of this work seem not to be very novel. Proposition 1 seems to be the most novel argument but I think it is problematic (see the later criticism). The Hungarian loss to compare sets and the skip connection are standard techniques.

* The proposed method suffers from high complexity. May I know how many rows of the incidence matrix I? Moreover, how to predefine the number of hyperedges/ the row # of I? The Hungarian loss has complexity O(#row(I)^2*#column(I)) or O(#row(I)*#column(I)^2) which could be high if we have a large number of rows or columns (nodes).

* My biggest concern is proposition 1. This is the most important technique argument. However, proposition 1 is not stated in a rigorous way at all. What is the definition of c? Can the authors highlight some intuition between the proof in the main text? I also checked the proof. It seems that the argument is not rigorous. There are quite a lot approximations, assumptions, etc. I do not even believe prop 1 is theoretically right.

* In experimental details, the authors should discuss how big the k can cover the datasets.


**Summary Of The Paper:**

This paper addressed two scaling problems in set-to-hypergraph prediction by pruning edges and gradients. The authors made an assumption that the maximum number of edges is k to reduce the memory complexity from O(2^n) to O(kn). Backpropagation with skips was also used to reduce the computation complexity. A series of experiments were done to show the method is empirically better than the baselines.

**Summary Of The Review:**

Though the set-to-hypergraph prediction is an important topic, the technical contributions of this paper are limited. The main statement is not rigorous and well explained.

---

### Author Response · Authors · 2021-11-16
**Response to all reviewers**

We thank all the reviewers for their thorough comments and their many suggestions for improving the exposition of the paper. We have updated the paper with the following main changes:
- We added section 1.1 Preliminary, which provides more explanations of the set-to-hypergraph task and an overview of Set2Graph (Serviansky et. al., 2020). Understanding the Set2Graph model is important for understanding the contributions of this work. This addresses concerns about accessibility for non-experts from reviewer ezwC and the question on what the input and output is from reviewer am1Q.
- We made Proposition 1 and its proof more rigorous, addressing the main concern by reviewer hMC3 and d3zQ. The new proposition includes details on how $J$ and $I$ are defined. Furthermore, we replaced the approximation in the proof with an inequality, addressing the concern by d3zQ on the correctness of the proof.
- We improved the exposition in Section 2, where the majority of the concerns from all the reviewers are located at. The changes include elaborations on the proposed methodology and highlight the necessity and soundness of pruning non-existing edges. Section 2 also includes the improved Proposition 1.

We will address the remaining comments in the second revision.

---

### Author Response · Authors · 2021-11-21
**2nd Response to all reviewers**

In our second revision to the paper, we addressed the remaining points by the reviewers, as we summarize here. We hope the reviewers will reconsider the scores in light of the improvements made in this and the previous revision. Please let us know if there is anything else that could be improved.

## On the number of edges [hMC3, d3zQ, am1Q]

The number of edges is pruned to a maximum number of $m$ (previously denoted as $k$) to improve the memory efficiency. In the experiments, we choose $m$ to be sufficiently large, but *minimal*, to cover all training examples. Specifically, this means that we set $m$ equal to the largest number of positive edges amongst any hypergraph in the training dataset.
Indeed, the number $m$ can be considered as a hyperparameter, with a greater $m$ improving the recall performance, but potentially at the cost of decreased precision. We choose a minimal $m$ since our goal is to improve the memory efficiency.

For the particle partitioning dataset $m=10$ edges suffice. For the convex hull finding task, the $m$ varies between different setups. The spherically distributed point sets have a constant number of edges that depends on the set size for $n\geq4$ as $m=(n-4)*2+4$. In the Gaussian distributed point sets, the number of edges are generally below 40 and almost all are below 60, even for $n=100$. We added the numbers and a discussion thereof to Appendix C.

## On the computational complexity [hMC3, d3zQ, ezwC]

Note that the time complexity of our loss computation is dominated by the algorithm that finds the minimizing assignment, which is in $\mathcal{O}(m^3)$. This is *independent* of the number of nodes (i.e., the input set size). In the discussion and future work section, we point out that this is currently the main computational bottleneck for problems with a large maximum number of edges. Furthermore, we include a discussion on the computational complexity in Appendix B.

## On training time [d3zQ, ezwC]

We mention the absolute training time of the experiments in Section 5.1 in the extended experiment discussion in Appendix C.
Note that the absolute training time of our proposed method is highly dependent on the type of hardware available, and hence can be very misleading when taken as an indicator for how long it will take on one's own hardware. For example, on a single GTX 1080 Ti and 10 cores of a 2.3GHz Xeon CPU, training takes under 12 hours for the particle partitioning experiment; on a single GTX 2080 Ti with 32 cores of a 3.7GHz Ryzen Threadripper CPU this time is cut by more than half.
In particular, it depends on the number of cores of the CPU and its clock speed. The CPU is necessary to find the optimal matching for the training loss. We discuss the computational complexity of this in Appendix B.

Figure 2 illustrates the trade-off between training time and predictive performance. In particular, it shows that backprop with skips offers a better F1 score at less training time. We also see from the results that the training time of our method is *not a fixed quantity*. In particular, we can improve the predictive performance of convex hull finding at the cost of longer training.


## On theoretical insights for the performance increase [ezwC, am1Q]

In section 5.2 we offer a quantitative study on how the number of recurrent steps affects the predictive performance on convex hull finding. In Figure 1, we show that when keeping this number fixed, the performance decreases with increasing input set sizes. Since we know the computational complexity of the problem ($\mathcal{O}(n\log(n)+n^{\text{floor}({\frac{d}{2}})})$ (Chazelle, 1993)), we also know that *any* solution for it will require computation time that scales super-linearly. Thus, we explain the drop in performance on larger set sizes as a result of the model *not having enough computation available*.

Based on this hypothesis, we tested whether adding more compute (by adding more iterations) *without increasing the number of parameters* can counteract the drop in performance that occurs due to increasing the set size.
Indeed, we see in Figure 1 that when increasing the number of recurrent steps, the performance remains roughly constant, despite an increase in the complexity of the problem. This is the main reason why our method drastically outperforms the Set2Graph baseline on convex hull finding and Delaunay triangulation for the larger input set sizes. We can expand the description of this in the Appendix if desired.

We thank the reviewers.

---

### Decision · Program_Chairs · 2022-01-20

**Decision:**

Reject

**Comment:**

This paper proposes techniques for improving the scalability of set-to-hypergraph models.
The main issue with the submission is that all reviewers found the clarity of the paper to be problematic including the proofs, the experimental conditions, and many other parts.
The authors responded but some reviewers explicitly state that their questions have only partially been answered and some reviewers did not respond to the authors. Unfortunately, given the number of clarity issues raised by the reviewers it makes more sense to re-submit this paper after re-writing based on all the suggestions from the reviewers.